# Re-Keying Scheme Revisited: Security Model and Instantiations

**Yuichi Komano** [1,] * and **Shoichi Hirose** [2]

1   Toshiba Corporation, Kawasaki 212-8582, Japan
2   Faculty of Engineering, University of Fukui, Fukui 910-8507, Japan; hrs_shch@u-fukui.ac.jp
*   Correspondence: yuichi1.komano@toshiba.co.jp; Tel.: +81-44-549-2156

**Abstract:**   The re-keying scheme is a variant of the symmetric encryption scheme where a sender (respectively, receiver) encrypts (respectively, decrypts) plaintext with a temporal session key derived from a master secret key and publicly-shared randomness. It is one of the system-level countermeasures against the side channel attacks (SCAs), which make attackers unable to collect enough power consumption traces for their analyses by updating the randomness (i.e., session key) frequently. In 2015, Dobraunig et al. proposed two kinds of re-keying schemes. The first one is a scheme without the beyond birthday security, which fixes the security vulnerability of the previous re-keying scheme of Medwed et al. Their second scheme is an abstract scheme with the beyond birthday security, which, as a black-box, consists of two functions; a re-keying function to generate a session key and a tweakable block cipher to encrypt plaintext. They assumed that the tweakable block cipher was ideal (namely, secure against the related key, chosen plaintext, and chosen ciphertext attacks) and proved the security of their scheme as a secure tweakable block cipher. In this paper, we revisit the re-keying scheme. The previous works did not discuss security in considering the SCA well. They just considered that the re-keying scheme was SCA resistant when the temporal session key was always refreshed with randomness. In this paper, we point out that such a discussion is insufficient by showing a concrete attack. We then introduce the definition of an SCA-resistant re-keying scheme, which captures the security against such an attack. We also give concrete schemes and discuss their security and applications.

**Keywords:** side channel attack; re-keying; tweakable block cipher; provable security

## 1. Introduction

Side channel attacks (SCAs) recover a secret key from a cryptographic device by collecting leakage information, such as the power consumption traces or the electro-magnetic traces, and by analyzing them statistically. Since the proposal by Kocher et al. [1], differential power analysis (DPA) has been one of the serious threats in the real world.

### 1.1. Background on Re-Keying Schemes

Against DPA, many countermeasures have been reported. At the device level, *masking* and *hiding* are studied well [2]. Masking is a countermeasure that randomizes the internal variables inside the module to disallow adversaries from analyzing the variables correctly. Hiding unlinks the internal values from the measured leakage to make the statistics meaningless. These countermeasures can be implemented

within the device itself and do not change the interface of the device. Therefore, they can be add-ons to existing systems.

On the other hand, as a system-level countermeasure, the re-keying scheme [3] has been proposed. It updates an encryption key frequently to make it infeasible for adversaries to collect leakage information. Medwed et al. [4] introduced the concept of "separation of duties" and proposed a concrete scheme for tiny devices. Their scheme consists of a re-keying function $F$ and a block cipher BC. Here, $F$ is a function that takes, as inputs, a master secret key $mk$ and a randomness $r$ to compute a temporal session key $tk$. They assumed that $F$ is easily protected from the SCAs. To encrypt a message $m$ with the length of the block size of BC, the scheme first chooses a randomness $r$ and computes the session key $tk$ by $F(mk, r)$. The scheme then encrypts $m$ with BC using the session key $tk$, without a countermeasure against the SCAs. Dobraunig et al. [5] provided an attack to recover the master secret key from Medwed et al.'s scheme [4]. In the next year, Dobraunig et al. [6] proposed two improvements. As for the first one, they reconsidered the property of $F$. In their attack against Medwed et al.'s scheme, they used the property that $F$ is invertible. To make the attack infeasible, they gave another example of $F$, which was non-invertible and pseudo-random in the ideal cipher model [7]. As for the second improvement, they gave a generic scheme replacing the block cipher of [4] with an ideal (secure against the related key attack in addition to the chosen cipher attack) tweakable block cipher to achieve the beyond birthday security. They showed its security by proving that the second scheme is a secure tweakable block cipher.

## 1.2. Our Contribution

This paper revisits the re-keying scheme. First, we point out that the previous works [4–6] did not give a formal security model. For example, Dobraunig et al. [6] discussed the security of their schemes with different security models. In addition, their models did not take the SCA resistance into consideration. In fact, as we give a concrete SCA attack, their model did not capture the security against the SCAs. Hence, we introduce a security model in considering the SCAs.

Second, we give two concrete first-order SCA-resistant re-keying schemes and discuss their security in our model. The first scheme is naturally derived from the combination of Dobraunig et al.'s second scheme [6] and Liskov et al.'s tweakable block cipher [8]. Unlike Dobraunig et al.'s abstract scheme, it is possible to discuss the SCA resistance with our concrete scheme. Our second scheme is a modification of the first scheme, which cannot be SCA resistant if it is used in the decryption device, which reveals the plaintext. However, it is useful for some applications in IoT systems, where the decrypted data are not revealed. We also discuss another scheme that is secure against the higher order SCA.

The paper is organized as follows. In the following subsection, we explain a related work recently reported. Section 2 reviews the definitions of the block cipher and the tweakable block cipher. In Section 3, we provide a message recovery attack with the SCA to the previous re-keying schemes. In considering such an attack, we introduce a new security model for the re-keying scheme in Section 4. We then give our concrete schemes and discuss their security in Sections 5 and 6, respectively. We also discuss the application of the re-keying scheme and the component of the re-keying function in Section 7. Finally, Section 8 concludes this paper.

## 1.3. Related Work

Dziembowski et al. [9] reconsidered the model of re-keying schemes and gave a new concrete scheme based on the hard physical learning problem. They first modeled the re-keying scheme, which assumed that the master secret key is stored as shares, and these shares were updated when the encryption/decryption process was invoked. Divide a secret key into randomized shares is one of the well-known countermeasures against the SCAs. Then, they gave two security models: against the black-box adversaries and the gray-box

ones. The first one denotes the security against adversaries who access the inputs and outputs of the re-keying scheme. On the other hand, the second one allows adversaries to see leakages of the scheme in addition to its inputs and outputs. They also proposed a concrete scheme based on the learning parity with leakage (LPL) problem, which can be reduced to the well-known problem, the learning parity with noise (LPN) problem.

Their models supposed the countermeasure using the shares. Moreover, their scheme is unsuitable for tiny devices since it requires a costly non-volatile write memory for updating shares. On the other hand, our schemes do not use the countermeasure with shares, which were not captured by their model. In addition, our schemes do not require such non-volatile write memory; hence, they are suitable for tiny devices.

Another device-level approach to enhance the security is also proposed. In [10], Chittamuru et al. proposed a framework that protects data from snooping attacks and improves hardware security. The re-keying scheme can be used as a module to improve the security of this framework.

## 2. Preliminaries

### 2.1. Block Cipher

The block cipher is a fundamental tool used for secure communication. It is also a building block of the re-keying scheme in the latter. Let us start with a review of a pseudo-random function, and then, we will review the definitions of the secure block cipher.

**Definition 1.** *Let* $\Phi$ *be a family of functions with* $\ell_n$*-bit input and* $\ell_m$*-bit output. We say that* $g(\cdot, \cdot)$ *is a pseudo-random function with* $\ell_n$*-bit input and* $\ell_m$*-bit output parameterized by an* $\ell_k$*-bit key k, if the advantage* Adv *below is negligible for any polynomial time adversary* $\mathcal{A}$ *who makes oracle queries to either* $g(k, \cdot)$ *or* $\varphi \in \Phi$ *up to q times:*

$$\text{Adv} := \max_{\mathcal{A}} |\Pr[\mathbf{Exp}_{g,\mathcal{A}}^{real} = 1] - \Pr[\mathbf{Exp}_{g,\mathcal{A}}^{rand} = 1]|,$$

*where experiments* $\mathbf{Exp}_{g,\mathcal{A}}^{real}$ *and* $\mathbf{Exp}_{g,\mathcal{A}}^{rand}$ *are as in Figure 1. In these experiments,* $\mathcal{O}_g(k, \cdot)$ *and* $\mathcal{O}_{\varphi}(\cdot)$ *are oracles, which, with* $\ell_n$*-bit input x, return* $g(k, x)$ *and* $\varphi(x)$*, respectively.*

| $\mathbf{Exp}_{g,\mathcal{A}}^{real}$: | $\mathbf{Exp}_{g,\mathcal{A}}^{rand}$: |
|---|---|
| $k \xleftarrow{\$} \{0,1\}^{\ell_k}$; | $\varphi \xleftarrow{\$} \Phi$; |
| return 1 iff $\mathcal{A}^{\mathcal{O}_g(k,\cdot)} = 1$ | return 1 iff $\mathcal{A}^{\mathcal{O}_\varphi(\cdot)} = 1$ |

**Figure 1.** Experiments for a pseudo-random function.

**Definition 2.** *Let* $\Pi$ *be a family of* $\ell_n$*-bit permutations. We say that* $(\mathsf{BC}, \mathsf{BC}^{-1})$ *is a pair of* $\ell_n$*-bit pseudo-random permutations parameterized by an* $\ell_k$*-bit key k, if the advantage* Adv *below is negligible for any polynomial time adversary* $\mathcal{A}$ *who makes oracle queries, with an* $\ell_n$*-bit input (plaintext), to either* $\mathsf{BC}$ *or* $\pi \in \Pi$ *up to q times.*

$$\text{Adv} := \max_{\mathcal{A}} |\Pr[\mathbf{Exp}_{\mathsf{BC},\mathcal{A}}^{real} = 1] - \Pr[\mathbf{Exp}_{\mathsf{BC},\mathcal{A}}^{rand} = 1]|$$

*where experiments* $\mathbf{Exp}_{\mathsf{BC},\mathcal{A}}^{real}$ *and* $\mathbf{Exp}_{\mathsf{BC},\mathcal{A}}^{rand}$ *are as in Figure 2. In these experiments,* $\mathcal{O}_{\mathsf{BC}}(k, \cdot)$ *and* $\mathcal{O}_\pi(\cdot)$ *are oracles, which, with* $\ell_n$*-bit input x, return the ciphertext* $BC(k, x)$ *and* $\pi(x)$*, respectively.*

Similarly, we say that $(\mathsf{BC}, \mathsf{BC}^{-1})$ is a strong pseudo-random permutation, if the advantage is negligible even if $\mathcal{A}$ allows making oracle queries, with an $\ell_n$ bit input (plaintext or ciphertext), to either $(\pi, \pi^{-1})$ or $(\mathsf{BC}, \mathsf{BC}^{-1})$, up to $q$ times in total.

| $\mathbf{Exp}_{\mathsf{BC},\mathcal{A}}^{real}$: | $\mathbf{Exp}_{\mathsf{BC},\mathcal{A}}^{rand}$: |
|---|---|
| $k \xleftarrow{\$} \{0,1\}^{\ell_k}$; | $\pi \xleftarrow{\$} \Pi(\cdot)$; |
| return 1 iff $\mathcal{A}^{\mathcal{O}_{\mathsf{BC}}(k,\cdot)} = 1$ | return 1 iff $\mathcal{A}^{\mathcal{O}_{\pi}(\cdot)} = 1$ |

**Figure 2.** Experiments for the block cipher.

**Definition 3.** *Let $\Pi$ be a family of $\ell_n$-bit permutations. We say that $(\mathsf{BC}, \mathsf{BC}^{-1})$ is a pair of $\ell_n$-bit pseudo-random permutations against related key attacks parameterized by an $\ell_k$-bit key $k$, if the advantage $\mathsf{Adv}$ below is negligible for any polynomial time adversary $\mathcal{A}$ who makes oracle queries to either $\mathcal{O}_{\mathsf{BC}}$ or $\mathcal{O}_{\Pi}$ up to $q$ times.*

$$\mathsf{Adv} \quad := \quad \max_{\mathcal{A}} | \Pr[\mathbf{Exp}_{\mathsf{BC},\mathcal{A}_{rk}}^{real} = 1] - \Pr[\mathbf{Exp}_{\mathsf{BC},\mathcal{A}_{rk}}^{rand} = 1]|$$

*where experiments $\mathbf{Exp}_{\mathsf{BC},\mathcal{A}_{rk}}^{real}, \mathbf{Exp}_{\mathsf{BC},\mathcal{A}_{rk}}^{rand}$ are as in Figure 3. Within them, $\mathcal{O}_{\mathsf{BC}}(k,\cdot,\cdot)$ and $\mathcal{O}_{\Pi}(l,\cdot,\cdot)$, given a difference $\Delta \in \{0,1\}^{\ell_k}$ and a plaintext $m$ from $\mathcal{A}_{rk}$, return $\mathcal{O}_{\mathsf{BC}}(k, \Delta, m) = \mathsf{BC}(k \oplus \Delta, m)$ and $\mathcal{O}_{\Pi}(l, \Delta, m) = \pi(m)$, respectively, where $\pi$ is chosen from $\Pi$ in accordance with $l \oplus \Delta$.*

*Similar to Definition 2, we have the notion of strong pseudo-random permutation against the related key attack.*

| $\mathbf{Exp}_{\mathsf{BC},\mathcal{A}_{rk}}^{real}$: | $\mathbf{Exp}_{\mathsf{BC},\mathcal{A}_{rk}}^{rand}$: |
|---|---|
| $k \xleftarrow{\$} \{0,1\}^{\ell_k}$; | $l \xleftarrow{\$} \{0,1\}^{\ell_k}$; |
| return 1 iff $\mathcal{A}_{rk}^{\mathcal{O}_{\mathsf{BC}}(k,\cdot,\cdot)} = 1$ | return 1 iff $\mathcal{A}_{rk}^{\mathcal{O}_{\Pi}(l,\cdot,\cdot)} = 1$ |

**Figure 3.** Experiments for the block cipher with related key attack.

## 2.2. Tweakable Block Cipher

The tweakable block cipher [8] is a variant of the block cipher. Besides the plaintext and the key, it takes an auxiliary input, called *a tweak*, which acts as an initial vector of the mode of operations [11]. By using different tweaks, ciphertexts differ even though the pair of the plaintext and the key is unique. This property makes the statistical analysis difficult.

Note that the tweak can be shared between a sender and a receiver in public. For example, the sender may select a tweak at random to send it with a ciphertext; or if both the sender and receiver are stateful and if they share a seed for tweaks, they synchronously compute a tweak without sending it. For simplicity, we assume that the sender sends a tweak along with a ciphertext over the public channel.

The tweakable block cipher consists of a pair of two algorithms $(\mathsf{TBC}, \mathsf{TBC}^{-1})$. The encryption algorithm $\mathsf{TBC}$ takes, as inputs, a key $k$, a tweak $t$, and a plaintext $m$ with bit lengths $\ell_k$, $\ell_t$, and $\ell_n$, respectively, to output an $\ell_n$-bit ciphertext $c$. The decryption algorithm $\mathsf{TBC}^{-1}$ takes, as inputs, $k$, $t$, and $c$ to recover $m$. They should satisfy the completeness; namely, $\mathsf{TBC}^{-1}(k, t, \mathsf{TBC}(k, t, m)) = m$ holds for arbitrary $k$, $t$, and $m$. We then review its security model.

**Definition 4.** *Let* $\Pi$ *be a family of* $\ell_n$*-bit permutations parameterized by an* $\ell_t$*-bit tweak.* $(\mathsf{TBC}, \mathsf{TBC}^{-1})$ *is a pair of* $\ell_n$*-bit pseudo-random permutations parameterized by an* $\ell_t$*-bit tweak and an* $\ell_k$*-bit key, if the advantage* Adv *below is negligible for any polynomial time adversary* $\mathcal{A}$ *who makes oracle queries, with a tweak and a plaintext, to either* TBC *or* $\Pi$ *up to q times.*

$$\mathsf{Adv} \quad := \quad \max_{\mathcal{A}} |\Pr[\mathbf{Exp}_{\mathsf{TBC},\mathcal{A}}^{real} = 1] - \Pr[\mathbf{Exp}_{\mathsf{TBC},\mathcal{A}}^{rand} = 1]|$$

*where experiments* $\mathbf{Exp}_{\mathsf{TBC},\mathcal{A}}^{real}$, $\mathbf{Exp}_{\mathsf{TBC},\mathcal{A}}^{rand}$ *are as in Figure* 4. *Within them,* $\mathcal{O}_{\mathsf{TBC}}(k, \cdot, \cdot)$ *and* $\mathcal{O}_\pi(\cdot, \cdot)$, *given a tweak* $t$ *and a plaintext* $m$ *from* $\mathcal{A}$, *return* $\mathsf{TBC}(k, t, m)$ *and* $\pi(t, m)$, *respectively.*

*Similar to Definition* 2, *we have the notion of strong pseudo-random permutation for the tweakable block cipher.*

Note that, unlike the key, the tweaks may be selected in an insecure manner (using the time information or sequential number, for example) and the adversary may control the manner. Especially, as for the (stateless) decryption, the adversary can make oracle queries with arbitrary tweaks of his/her choice. Therefore, the above definitions assume the adversary in the related tweak attack setting, i.e., the open-tweak model [12]. Moreover, similar to the block cipher, the security model can be extended against the related key attacks. The model is similar to Definition 3, and we omit it here.

| $\mathbf{Exp}_{\mathsf{TBC},\mathcal{A}}^{real}$: | $\mathbf{Exp}_{\mathsf{TBC},\mathcal{A}}^{rand}$: |
|---|---|
| $k \xleftarrow{\$} \{0,1\}^{\ell_k}$; | $\pi \xleftarrow{\$} \Pi(\cdot, \cdot)$; |
| return 1 iff $\mathcal{A}^{\mathcal{O}_{\mathsf{TBC}}(k,\cdot,\cdot)} = 1$ | return 1 iff $\mathcal{A}^{\mathcal{O}_\pi(\cdot,\cdot)} = 1$ |

**Figure 4.** Experiments for the tweakable block cipher.

## 3. SCA on the Previous Re-Keying Schemes

In this section, we review the previous re-keying schemes; Medwed et al.'s re-keying scheme and Dobraunig et al.'s first re-keying scheme. We then show the plaintext recovery attacks with the SCA against these schemes.

### 3.1. Previous Works

Medwed et al. [4] introduced the design concept of "separation of duties" and proposed a concrete re-keying scheme suitable for lightweight devices. Their scheme consists of two parts as in Figure 5. The first part is a re-keying function $F$, which takes a master secret key $mk$ and randomness $r$ as inputs and outputs a session key $tk$. The re-keying function is designed with simple operations, which is easily protected from the SCAs. The second part is a block cipher $BC$, which encrypts a plaintext $m$ with the session key $tk$. Decryption consists of the above $F$ and $BC^{-1}$, which is the inverse of $BC$. They assumed that $F$ was multiplicative; precisely, $tk = mk \cdot r$ in a finite field. Against their scheme, Dobraunig et al. [5] showed an attack that first searches $tk$ with the birthday attack and then recovers $mk$ by $mk = tk \cdot r^{-1}$.

Dobraunig et al. [6] then gave two other schemes and discussed their security. Figure 6 depicts their first scheme. Their first scheme is an improvement of Medwed et al.'s scheme by replacing the re-keying function with a non-invertible function. Figure 6 depicts their construction, consisting of an SCA-resistant function $g$ and a one-way function $h$.

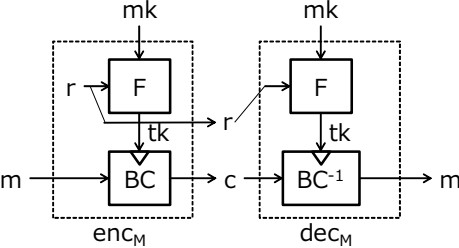

**Figure 5.** Medwed et al.'s scheme.

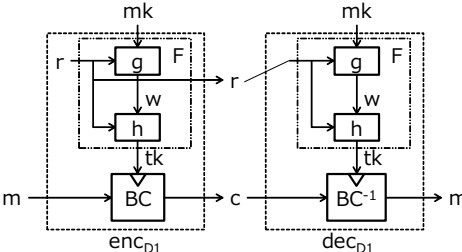

**Figure 6.** Dobraunig et al.'s first scheme.

*3.2. The Attacks*

The previous works supposed that the re-keying schemes are secure if the randomness *r* is fresh, and they discussed the security against the SCAs insufficiently. In fact, as they supposed, *the key recovery attack* on both the master secret key *mk* and the session key *tk* seems infeasible with the SCA if the randomness is fresh, because of the property (assumption) of *g*.

Let us discuss another attack, a message recovery attack, with the SCAs. Assume an adversary who, monitoring a ciphertext $(r, c)$, aims to recover the plaintext *m* corresponding to *c*. Against Medwed et al.'s re-keying scheme and Dobraunig et al.'s first scheme, such an attack is feasible if the decryption device accepts decrypting an arbitrary input without checking the freshness of *r*. The attack proceeds as follows. The adversary repeats sending a decryption query $(r, c')$ for randomly-chosen $c'$. The attacker measures the side channel information. For the fixed randomness *r*, the session key *tk* is also fixed; and therefore, the adversary can determine *tk* with the SCA. Once the session key *tk* is determined, the adversary can recover *m* by decrypting *c* with *tk*. The assumption, where the decryption device does not check the freshness of *r*, is realistic because Medwed et al. assumed stateless devices as the lightweight devices.

## 4. Security Model of the Re-Keying Scheme in Considering the SCA

In this section, we introduce a security model considering the SCA. Except the SCA resistance, the model is similar to that of the tweakable block cipher. Namely, the re-keying scheme is secure if it is indistinguishable from a random function. To take the SCA resistance into consideration, we assume that the encryption and decryption oracles leak the side channel information.

Unlike the tweakable block cipher, we assume that the encryption device chooses a randomness (which corresponds to the tweak in the tweakable block cipher) uniformly random for each encryption. Hence, we disallow the adversary from choosing it in the encryption oracle query. On the other hand, the decryption device receives the randomness as one of the inputs; therefore, we allow the adversary to choose it in the decryption query.

As for the side channel information, we introduce a leakage function $\mathcal{L}(\text{BC})$, which returns the leakages (e.g., a power consumption trace) through the block cipher operation. In the real world, the SCAs on the key loading and XORing with the key have been reported. However, there are protection mechanisms to decrease the platform leakage (e.g., for the key loading), and the leakage is small in XORing compared to the complex operations in the block cipher. Hence, we omit them. Moreover, we assume that the re-keying function is properly designed not to leak the side channel information of the key.

**Definition 5.** *Let $\Pi$ be a family of $\ell_n$-bit permutations parameterized by an $\ell_r$-bit randomness. $(E, E^{-1})$ is a pair of $\ell_n$-bit pseudo-random permutations parameterized by an $\ell_r$-bit randomness and an $\ell_k$-bit key, if the advantage* Adv *below is negligible for any polynomial time adversary $\mathcal{A}$ who makes oracle queries, with a plaintext, to either E or $\Pi$ up to q times.*

$$\text{Adv} \quad := \quad \max_{\mathcal{A}} |\Pr[\mathbf{Exp}_{\text{RK},\mathcal{A}}^{real} = 1] - \Pr[\mathbf{Exp}_{\text{RK},\mathcal{A}}^{rand} = 1]|$$

*where experiments $\mathbf{Exp}_{\text{RK},\mathcal{A}}^{real}$, $\mathbf{Exp}_{\text{RK},\mathcal{A}}^{rand}$ are as in Figure 7. Within them, $\mathcal{O}_{\text{RK}}(k, R, \cdot)$, given a plaintext m from $\mathcal{A}$, returns both $\text{RK}(k, r, m)$ for a fresh randomness r chosen by the oracle itself and $\mathcal{L}(\text{BC}(tk, m))$, where tk is a session key derived from the re-keying function with k and r. On the other hand, $\mathcal{O}_\Pi(R, \cdot)$, given a plaintext m from $\mathcal{A}$, returns both $\pi(r, m)$ for a fresh randomness r chosen by the oracle itself and $\mathcal{L}(\text{BC}(tk, m'))$, where tk and $m'$ are a session key derived from the re-keying function with k and r and an $\ell_n$-bit random plaintext chosen by the oracle itself, respectively.*

*Similar to Definition 2, we have the notion of strong pseudo-random permutation for the re-keying scheme. In this case, note that the adversary is allowed to choose the randomness r as an input of the decryption oracle.*

| $\mathbf{Exp}_{\text{RK},\mathcal{A}}^{real}$: | $\mathbf{Exp}_{\text{RK},\mathcal{A}}^{rand}$: |
|---|---|
| $k \xleftarrow{\$} \{0,1\}^{\ell_k}$; | $\pi \xleftarrow{\$} \Pi(\cdot, \cdot)$; |
| return 1 iff $\mathcal{A}^{\mathcal{O}_{\text{RK}}(k,R,\cdot)} = 1$ | return 1 iff $\mathcal{A}^{\mathcal{O}_\pi(R,\cdot)} = 1$ |

**Figure 7.** Experiments for the re-keying scheme.

## 5. New Concrete Re-Keying Schemes

We first recall our building blocks: Liskov et al.'s tweakable block cipher and Dobraunig et al.'s second re-keying schemes. We then give our concrete re-keying schemes.

### 5.1. Building Blocks

Liskov et al. [8] introduced the concept of the tweakable block cipher and gave several schemes. One of the schemes, known as LRW1, calls the block cipher with a secret key $k$ twice as inner components where the input of one block cipher is the XOR of the tweak and the output of the other block cipher. Its latency is about twice as one of the block cipher's. Liskov et al. proved that LRW1 is a pseudo-random permutation.

The other scheme, LRW2, described in Figure 8, reduces the number of the block cipher operations by one. It, however, requires additional operation of a keyed hash function. The output of the hash function is XORed with both the input and output of the block cipher. If an encryption device computes the keyed hash function in advance and stores the output, its latency is half of LRW1's. Liskov et al. proved that LRW2 is a strong pseudo-random permutation, if $h_l$ is chosen from the $\delta_h$-AXU$_2$ hash function family $h_L$. Here, $h_L$ is the $\delta_h$-AXU$_2$ ($\delta_h$-almost two-XOR-universal [8]) hash function family if $\Pr_l[h_l(x) \oplus h_l(y) = z] \leq \delta_h$ holds for all $x, y$, and $z$ such that $x \neq y$.

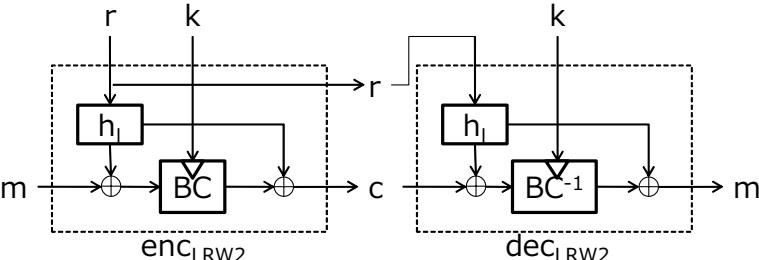

**Figure 8.** Liskov et al.'s tweakable block cipher: LRW2.

Dobraunig et al. [6] followed the concept of "separation of duties" to propose two other re-keying schemes. Their first scheme is identical to Medwed et al.'s scheme, but the requirement of $F$ is different. They added the one-wayness property (precisely, Dobraunig et al. divided $F$ into two parts: the first and second parts are assumed to have SCA resistance and one-wayness, respectively) to $F$ and proved its security in the ideal cipher model. Note that their first scheme avoids the attack to recover $mk$ from $tk$; however, it is possible to search for the session key $tk$ by the birthday attack.

Their second scheme, depicted in Figure 9, replaces BC with a tweakable block cipher to achieve the beyond birthday security. The birthday attack above succeeds because the re-keying function and the block cipher work independently, and adversaries are able to collect the input-output pairs of the block cipher and the re-keying scheme, step by step. Their second scheme, however, binds the re-keying function with the (tweakable) block cipher by inputting the randomness into the tweakable block cipher as a tweak. It makes it difficult for adversaries to collect the input-output pairs independently.

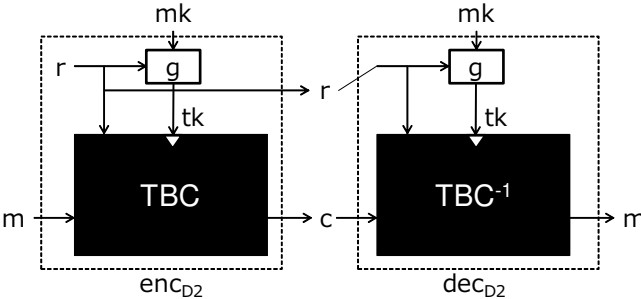

**Figure 9.** Dobraunig et al.'s second re-keying scheme (generic construction).

*5.2. Our Concrete Schemes*

5.2.1. First-Order SCA-Resistant Re-Keying Scheme

Our first scheme is based on the combination of LRW2 and Dobraunig et al.'s second scheme. From Theorem 3 of [6], the combination of these schemes is a concrete re-keying scheme; however, the keyed hash function in LRW2 may leak the side channel information. Hence, we modified it as depicted in Figure 10 to achieve the security against the first-order SCAs.

This scheme uses three functions. The first function is a re-keying function $g_1$, which, given the master secret key $mk_1$ and the randomness $r$, returns the session key $tk$. We assume that $g_1$ is a pseudo-random permutation when one of $mk_1$ and $r$ is fixed and that $g_1$ leaks no side channel information on inputs. The second function is a pseudo-random function $g_2$, which, given another master secret key $mk_2$ and

the randomness $r$, returns the $\ell_n$-bit string $n$. We assume that $g_2$ also leaks no side channel information on the inputs. The third function is a block cipher BC. As discussed later, the re-keying scheme is secure against the first-order SCA because the input and output of BC are XORed (masked, as in the masking countermeasure) by $n$, which is unknown to the adversary. The procedures of this scheme $(\text{enc}_1, \text{dec}_1)$ are described in Figure 10 and Algorithms 1 and 2.

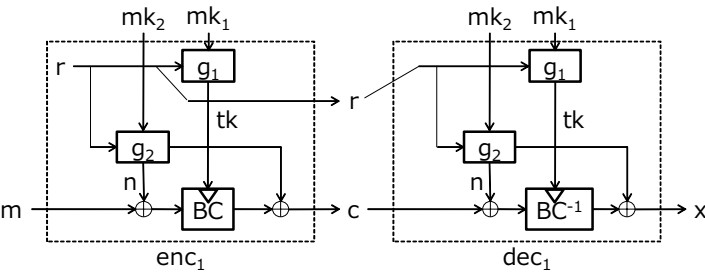

**Figure 10.** First-order SCA-resistant re-keying scheme.

---

**Algorithm 1** First-order SCA-resistant re-keying scheme: $\text{enc}_1$

---

**Input:** master secret key $(mk_1, mk_2)$ and plaintext $m$
**Output:** ciphertext $(r, c)$

---

1. Choose a randomness $r$
2. Compute $tk = g_1(mk_1, r)$
3. Compute $n = g_2(mk_2, r)$
4. Compute $c = \text{BC}(tk, m \oplus n) \oplus n$
5. Return $(r, c)$

---

**Algorithm 2** First-order SCA-resistant re-keying scheme: $\text{dec}_1$

---

**Input:** master secret key $(mk_1, mk_2)$ and ciphertext $(r, c)$
**Output:** plaintext $m$

---

1. Compute $tk = g_1(mk_1, r)$
2. Compute $n = g_2(mk_2, r)$
3. Compute $m = \text{BC}^{-1}(tk, c \oplus n) \oplus n$
4. return $m$

---

### 5.2.2. SCA-Resistant Re-Keying Encryption Scheme

Our second scheme is a modification of the first scheme, by removing the XOR operation before the block cipher operation. The second scheme lacks the resistance against the message recovery attack with the SCA, if it is used in the decryption device as follows. Assume that an SCA adversary, given a target $(r, c)$, is allowed to use the decryption device as an oracle. The attacker queries $(r, c')$ for randomly-chosen $c'$ and receives $m'$ with the leakage $\mathcal{L}(\text{BC}^{-1}(tk, c))$ where $tk = g_1(mk_1, r)$. Although the input of $\text{BC}^{-1}$ is protected with a mask $g_2(mk_2, r)$, the output is unprotected; and hence, the adversary can recover $tk$ with the first-order SCA and decrypt $m$ with the recovered $tk$.

Note that, if the decryption device does not output the decrypted data, i.e., the decryption oracle returns only $\mathcal{L}(\text{BC}^{-1}(g_1(mk_1, r), c))$ without $m$, the above message recovery attack with the SCA is infeasible.

Furthermore, note that if the encryption device (i.e., the encryption oracle) chooses a randomness $r$ properly, the encryption device is secure against the first-order SCA without masking the input of BC, since the encryption key $tk$ is a fresh randomness. The procedures of this scheme $(\text{enc}_2, \text{dec}_2)$ are described in Figure 11 and Algorithms 3 and 4.

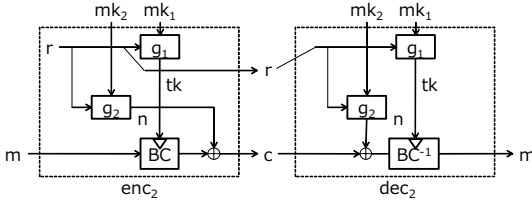

**Figure 11.** First-order SCA-resistant re-keying encryption scheme.

---

**Algorithm 3** First-order SCA-resistant re-keying encryption scheme: $\text{enc}_2$

---

**Input:** master secret key $(mk_1, mk_2)$ and plaintext $m$
**Output:** ciphertext $(r, c)$

---

1. Choose a randomness $r$
2. Compute $tk = g_1(mk_1, r)$
3. Compute $n = g_2(mk_2, r)$
4. Compute $c = \text{BC}(tk, m) \oplus n$
5. Return $(r, c)$

---

**Algorithm 4** First-order SCA-resistant re-keying encryption scheme: $\text{dec}_2$

---

**Input:** master secret key $(mk_1, mk_2)$ and ciphertext $(r, c)$
**Output:** plaintext $m$

---

1. Compute $tk = g_1(mk_1, r)$
2. Compute $n = g_2(mk_2, r)$
3. Compute $m = \text{BC}^{-1}(tk, c \oplus n)$
4. Return $m$

---

## 6. Security Considerations

In this section, let us discuss the security of our schemes. For each scheme, we first show that the adversary has no meaningful information from the SCA. Then, we show that the adversary's advantage for the distinguishing game in Definition 5 is negligible.

### 6.1. Security of the SCA-Resistant Re-Keying Scheme

In the scheme in Figure 10, the input and the output of BC are masked with $g_2(mk_2, r)$, which is unknown to the adversary. Hence, the adversary cannot guess the internal value of BC, nor retrieve the meaningful side channel information. Therefore, it is obvious that the scheme is resistant against the first-order SCA.

The scheme is a modification of the combinatorial scheme of LRW2 and Dobraunig et al.'s second scheme, restricting the keyed hash function in LRW2 not to leak the side channel information. The security model of the re-keying scheme in Definition 5 is a subset of the tweakable block cipher in Definition 4. From Theorem 3 of [6] and the above discussions, the scheme naturally satisfies Definition 5. More precisely, we have the following theorem for the scheme.

**Theorem 1.** *Assume that* BC *is a (strong) pseudo-random permutation and that $g_1$ is a pseudo-random permutation. Then, the scheme of Figure 10 is a (strong) pseudo-random re-keying scheme such that:*

$$\mathsf{Adv} \leq \epsilon_{g_1} + q\epsilon_B,$$

*where $\epsilon_B$ and $\epsilon_{g_1}$ are upper bounds on the advantage of the adversaries against* BC *and $g_1$, respectively.*

**Proof.** For simplicity, this proof is for a pseudo-random permutation BC. The proof for a strong pseudo-random permutation BC is very similar.

Let $\mathcal{A}$ be an adversary against the re-keying scheme. $\mathcal{A}$ has oracle access to $\mathcal{O}_{\mathsf{RK}}$.

Let $\mathcal{D}_1$ be an adversary against $g_1$. $\mathcal{D}_1$ runs $\mathcal{A}$ and simulates $\mathcal{O}_{\mathsf{RK}}$ for $\mathcal{A}$. Then,

$$\Pr[\mathbf{Exp}_{\mathsf{RK},\mathcal{A}}^{real} = 1] = \Pr[\mathbf{Exp}_{g_1,\mathcal{D}_1}^{real} = 1] \leq \epsilon_{g_1} + \Pr[\mathbf{Exp}_{g_1,\mathcal{D}_1}^{rand} = 1].$$

Let $\mathcal{D}_2$ be an adversary against BC. $\mathcal{D}_2$ is given $q$ different oracles, which are either $(\mathsf{BC}(k_1,\cdot),\ldots,\mathsf{BC}(k_q,\cdot))$ or $(\pi_1(\cdot),\ldots,\pi_q(\cdot))$, where each $k_i$ is chosen uniformly at random from $\{0,1\}^{\ell_k}$ and each $\pi_i$ is chosen uniformly at random from the set of all permutations over $\{0,1\}^{\ell_n}$. $\mathcal{D}_2$ runs $\mathcal{A}$ and simulates $\mathcal{O}_{\mathsf{RK}}$ for $\mathcal{A}$. For a query $(r,m)$ made by $\mathcal{A}$, if there exists a previous query $(r',m')$ such that $r = r'$, then $\mathcal{D}_2$ asks $m$ from the oracle from which $\mathcal{D}_2$ asked $m'$. Otherwise, $\mathcal{D}_2$ asks $m$ from a new oracle. Then,

$$\Pr[\mathbf{Exp}_{g_1,\mathcal{D}_1}^{rand} = 1] = \Pr[\mathcal{D}_2^{\mathsf{BC}(k_1,\cdot),\ldots,\mathsf{BC}(k_q,\cdot)} = 1] \leq q\epsilon_B + \Pr[\mathcal{D}_2^{\pi_1(\cdot),\ldots,\pi_q(\cdot)} = 1].$$

It is not difficult to see that:

$$\Pr[\mathcal{D}_2^{\pi_1(\cdot),\ldots,\pi_q(\cdot)} = 1] = \Pr[\mathbf{Exp}_{\mathsf{RK},\mathcal{A}}^{rand} = 1].$$

Thus,

$$\left|\Pr[\mathbf{Exp}_{\mathsf{RK},\mathcal{A}}^{real} = 1] - \Pr[\mathbf{Exp}_{\mathsf{RK},\mathcal{A}}^{rand} = 1]\right| \leq \epsilon_{g_1} + q\epsilon_B.$$

□

The above theorem holds whether BC is secure against the related key attack or not. Although Dobraunig et al.'s second scheme requires the ideal tweakable block cipher (namely, secure against the related key attack), our scheme can relax the requirement for its building block.

*6.2. Security of the SCA-Resistant Re-Keying Encryption Scheme*

Let us discuss the security of the scheme in Figure 11. Assume that an SCA adversary mounts the attacks on the encryption device, whereas the SCA on the decryption is restricted. Similar to the previous scheme, this scheme is also SCA resistant if it is used in the encryption device.

Let us discuss the SCA resistance of the decryption device. Without restriction, it is vulnerable to the message recovery attack with the SCA, because the output of $\mathsf{BC}^{-1}$ is unprotected. If we restrict the adversary not to receive $m$, which is the output of $\mathsf{BC}^{-1}$, the attack is difficult only with the leakage information.

The security of our second scheme, by ignoring the side channel information, can be proven in a similar way as Theorem 1.

### 6.3. Toward the Higher Oder SCA-Resistant Schemes

The schemes in Figures 10 and 11 are secure against the first-order SCAs. To resist the higher order SCA, we need more random freshnesses in general. Assume that, in addition to $r$, the randomnesses $r_2, r_3, \cdots$ are randomly generated in the encryption device to compute $n_2, n_3, \cdots$, which mask the input and the output of BC, and sent to the decryption device as elements of $(r, r_2, r_3, \cdots, c)$. It seems to lead the higher order SCA-resistant scheme. However, it is insecure because the message recovery attack, by reusing $(r, r_2, r_3, \cdots)$, is possible.

Let us consider another scheme in Figure 12 and Algorithms 5 and 6. The idea to achieve the higher order SCA resistance is to generate a fresh randomness for a pair $(m, r)$. In the SCA, the adversary queries $m$ or $(r, c)$ as the encryption query or the decryption query, respectively. By letting the mask depend on the pair, the masks for the input and the output of BC should be always fresh for each query. This prevents the higher order SCA.

The scheme in Figure 12 is complicated, unlike the previous two schemes, which shows the trade-off between the security and efficiency. Another construction, where the session key also depends on both $(r, m)$, can be considered.

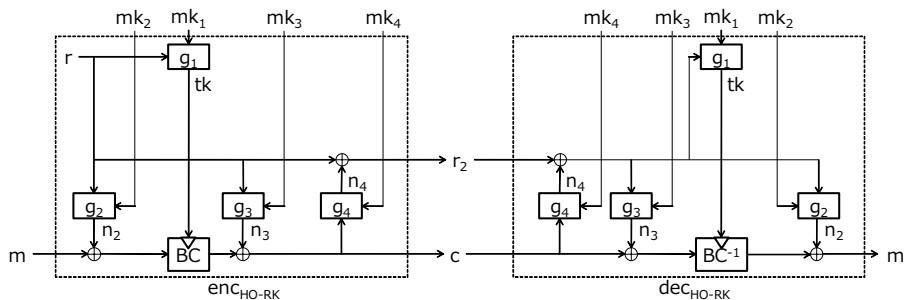

**Figure 12.** Higher order SCA-resistant re-keying scheme.

---

**Algorithm 5** Higher order SCA-resistant re-keying encryption scheme: $\mathsf{enc}_{\mathrm{HO-RK}}$

---

**Input:** master secret key $(mk_1, mk_2, mk_3)$ and plaintext $m$
**Output:** ciphertext $(r, c)$

---

1. Choose a randomness $r$
2. Compute $tk = g_1(mk_1, r)$
3. Compute $n_2 = g_2(mk_2, r)$
4. Compute $c = \mathsf{BC}(tk, m \oplus n_2) \oplus g_3(mk_3, r)$
5. Compute $r_2 = r \oplus g_4(mk_4, c)$
6. Return $(r_2, c)$

---

**Algorithm 6** Higher-order SCA-resistant re-keying encryption scheme: $\mathsf{dec}_{\mathrm{HO-RK}}$

---

**Input:** master secret key $(mk_1, mk_2, mk_3)$ and ciphertext $(r_2, c)$
**Output:** plaintext $m$

---

1. Compute $r = r_2 \oplus g_4(mk_4, c)$
2. Compute $n_4 = g_4(mk_4, r)$
3. Compute $tk = g_1(mk_1, r)$
4. Compute $m = \mathsf{BC}^{-1}(tk, c \oplus n_4) \oplus g_2(mk_2, r)$
5. Return $m$

---

## 7. Discussion

The following subsections discuss two applications for the first-order SCA-resistant encryption re-keying scheme and the first-order SCA-resistant decryption re-keying scheme.

### 7.1. Application to Sensor Network Devices

Let us assume sensor devices that collect the sensor data and send them to a server via an edge device. The sensor devices and the edge device are supposed to be resource-constrained. Since these devices are located in the field, there is a fear of the SCA on these devices.

Furthermore, assume that the sensor data are confidential. For example, the sensor devices are medical ones, and all the data include sensitive information of the patients. Alternatively, the sensor devices are located at an agricultural field, and all the data include key information, such as the temperature or the humidity, which is useful for optimal cultivation. Furthermore, let us assume that the data sent from the server to the devices are less sensitive; for example, the instructions for collecting and sending sensor data. Hence, let us consider that the sensor device encrypts its sensor data and sends them to the edge device; and then, the edge device decrypts the sensor data, re-encrypts them with a key shared between the edge device and the server, and sends the ciphertext to the server. Note that the decrypted sensor data are not revealed by the edge device.

The first-order SCA-resistant encryption re-keying scheme is suitable for such a situation. Let us regard the sensor devices and the edge device as the encryption device and the decryption one in the scheme, respectively. As we discussed in Section 5.2.2, the SCA against the sensor device is difficult because of the fresh randomness $r$. As for the edge device, it does not reveal the decrypted sensor data, but the re-encrypted ciphertext; therefore, the SCA against the edge device is also difficult, as we discussed in Section 5.2.2.

Note that the previous schemes, Medwed et al.'s scheme and Dobraunig et al.'s first scheme, are unsuitable for this application. This is because, in these schemes, the input of $\mathrm{BC}^{-1}$ in the edge device is unprotected and the message recovery attack with the SCA against the edge device is possible.

### 7.2. Construction of g

If $g$ is resistant to the SCAs, our schemes are theoretically resistant to the SCAs. Therefore, our aim is to construct $g$, which consists of operations with less side channel leakage and/or easily-added countermeasures against the SCAs.

Generally speaking, the non-linear operations, such as SubByte in AES [13], tend to leak the side channel information. The implementations of these operations tend to be complicated circuits, which require much power consumption, including the meaningful information. Moreover, from the implementability, the inputs and outputs of these operations are restricted by a small bit length. This enables adversaries to guess the inputs and outputs to succeed in the statistics of the SCAs. Therefore, it is better to construct the re-keying function with the simple (linear) operations, rather than the non-linear one, to mix the master secret key and the randomness.

Note that Theorem 1 requires that $g$ is a (strong) pseudo-random permutation; namely, it is a one-to-one pseudo-random function if one of $mk$ and $r$ is fixed. An example of such $g$ is a composition of an SCA-free function $g_{\mathsf{MIX}}$ and a pseudo-random permutation $g_{\mathsf{PRP}}$. The SCA-free function is, for example, $g_{\mathsf{MIX}}(mk, r) = mk \cdot T \oplus r$, where $T$ is a regular $128 \times 128$ matrix consisting of zero or one, and $mk \cdot T$ is a multiplication of a $128 \times 1$ vector $mk$ and $T$. Other examples can be obtained by dividing the above $T$ into $64 \times 64$ or $32 \times 32$ matrices. In addition, the masking countermeasures are easily applicable to linear functions including such $g_{\mathsf{MIX}}$.

## 8. Conclusions

In this paper, we reconsidered the re-keying scheme. We pointed out that the previous works lacked the security consideration on the SCA and introduced a security model of the re-keying scheme considering the SCA. We then gave concrete schemes and discussed their security and applications. In the IoT era, whereas the SCAs are a serious threat to the IoT devices, these devices are resource-constrained, and it is difficult to implement the existing countermeasures on them. In addition to the device-level approach, systematic countermeasures such as the re-keying scheme are some of the promising countermeasures.

**Author Contributions:** conceptualization, Y.K.; methodology, Y.K. and S.H.; formal analysis, Y.K. and S.H.; writing—original draft preparation, Y.K.; writing—review and editing, Y.K. and S.H.

**Funding:** This work was supported by JSPS KAKENHI Grant Number JP18H05289.

**Conflicts of Interest:** The authors declare no conflict of interest.

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
