# Peer review of "Re-Keying Scheme Revisited: Security Model and Instantiations"

_applsci, doi:10.3390/app9051002_

Round 1

Reviewer 1 Report

In this paper re-keying schemes are revisited and new insights are presented which provide solutions to the previous published re-keying schemes under SCA attacks. The paper is clearly presented and well organized but it has several points which need correction/improvement. Below I do a chronology of them as I have found during reading.

1) Figure 5 (right): it is uncomplete.

2) Definition 4: could you explicitly define operations (E,D)?

3) Figure 8: it is uncomplete.

4) Espression after line 298: subindexs are wrong.

5) Line 304: there are mistakes in parentheses.

The scheme presented in Figure 12 needs further explanation.

6) Figure 12: What is the role of r2? It is unclear.

7) Figure 12: what is the role of g6? Is is unclear.

8) Figure 12: The resistance of g3 to SCA attacks is unclear to me. Despite g functions are declared as SCA resistant, they always receive a fresh random r, plus a master key. In this particular case, function g3 receives a master key mk3 and plaintext (totaly under control of the attacker). I cannot imagine how g3 could resist a SCA attack under these conditions. Please explain.

9) Line 329: correct mistake in 2g.

10) Line 340: bold text out of context.

11) There are references not cited in the text: [8,13-19]

Author Response

Reply Letter

Dear Editor and Reviewers,

We sincerely appreciate your invaluable comments on our manuscript. We revised it in considering your comments as follows.

In this revision, we removed subsections on our third scheme (re-keying decryption scheme) because there is a security flaw in the scheme, against the message recovery attack with the SCA. We also changed the higher-order SCA resistant re-keying scheme to be more simple.

We added our reply in-line with the red color.

Thank you again and best regards,

Yuichi Komano and Shoichi Hirose

Reviewer 1

Comments and Suggestions for Authors

1) Figure 5 (right): it is uncomplete.

Figure 5 right (in the initial submission) showed a concrete construction by replacing h in Fig. 5 (left) with another BC and an XOR. However, our attack in Section 3.2 generally works against the generic construction (Fig. 5, left). Hence, we removed the figure of Fig. 5 (right) in this revision.

We also added a figure corresponding to the decryption, in order to make the procedures of encryption and decryption clear. We modified other figures in the similar reason.

2) Definition 4: could you explicitly define operations (E,D)?

We replaced (E,D) with (E,E^{-1}) and added explanations, in order to clarify that D is inverse of E.

3) Figure 8: it is uncomplete.

Dobraunig et al.’s second scheme is a generic construction which uses a TBC encryption scheme as a black-box. They did not specify the construction of the TBC. To make it clear, we changed the box with the black color.

4) Espression after line 298: subindexs are wrong.

We modified the sub-index from g and h to g_1 and g_2, respectively.

5) Line 304: there are mistakes in parentheses.

We entirely modified the proof. Since the new proof is generic, we moved the theorem and the proof to the previous subsection.

The scheme presented in Figure 12 needs further explanation.

6) Figure 12: What is the role of r2? It is unclear.

We updated the scheme. Similar to the old scheme in the initial submission, there is r2 which is required for the decryption. We added the diagram of decryption in each figure, in order to make the decryption process clear.

7) Figure 12: what is the role of g6? Is is unclear.

The functions g6 in the old scheme and g4 in the new scheme are required in order for the masks for the input/output of BC/BC-1 to be dependent on both r and c.

The higher-order SCA makes the first-order masking meaningless. Against our first-order SCA resistant schemes, the higher-order SCA adversary, aiming to decrypt a message for a target ciphertext (r*,c*), makes queries (r*,c_i) for fixed r* and random c_i. With these queries, the adversary can reveal the session key tk with the higher-order SCA.

With g6 and g4, the masks for the input/output of BC/BC-1 are unique (different) for each (r*,c_i), even though r* is fixed, which leads the higher-order SCA resistant.

8) Figure 12: The resistance of g3 to SCA attacks is unclear to me. Despite g functions are declared as SCA resistant, they always receive a fresh random r, plus a master key. In this particular case, function g3 receives a master key mk3 and plaintext (totaly under control of the attacker). I cannot imagine how g3 could resist a SCA attack under these conditions. Please explain.

We removed g3 in the new scheme in the revision.

9) Line 329: correct mistake in 2g.

We changed the proof entirely as we explained above.

10) Line 340: bold text out of context.

We modified \mathbf{BC} to \mathsf{BC}.

11) There are references not cited in the text: [8,13-19]

We re-compiled to remove the un-referred ones.

Thank you very much again for your valuable comments.

Reviewer 2

Comments and Suggestions for Authors

Big items:

-It is recommended to add experimental section in this paper which compares proposed security enhancement scheme with previously proposed re-keying mechanisms to appreciate authors contribution.

The re-keying scheme theoretically ensures the SCA resistance. It depends on the construction of re-keying function, especially $g$. $g$ should be well designed not to leak the information of the master secret key for each platform (i.e., for each CPU, hardware libraries, etc.). We added a discussion on it in Section 7.2.

-Section 7: This section appears concise, it is recommended to quantify the overheads of the proposed rekeying scheme for these applications. In addition, it is even better compare and contrast this scheme overhead with other schemes as well.  

The overhead depends on the construction of $g$ as we discussed in Section 7.1.

On the comparison with previous schemes, especially, with the Medwed et al.’s scheme and the Dobraunig et al.’s first scheme, the comparison seems meaningless because these schemes are insecure for this application. We added a discussion on the security with these schemes in Section 7.1. On the other hand, since the Dobraunig et al.’s second scheme is not concrete, we cannot discuss the comparison. Hence, we cannot give the comparison.

-Throughout this paper there are many single line paragraphs. To improve formatting of the paper, my suggestion is to either make these single line sentences into subheadings or merge them to next paragraph.      

We changed the sentences to reduce the single-sentence paragraph.

-In Section 5.2, encryption and decryption schemes are presented as bullets. In current form these bullets are not clearly understandable. It is recommended to convert these bullets into algorithms such that a reader understand these schemes clearly.     

We converted the bullets to the algorithms below the diagram, in order to understand the procedure easily.

Small Things:

-Page 2 of 13: Section 1.3: This section starts with a single sentence paragraph, which is not recommended. It is suggested to merge this paragraph to next paragraph.

We merged the first and second paragraphs as you suggested.

-Page 2 of 13: Section 1.3, the literature survey in this section can be improved. As this paper presents keying mechanisms at both device- and system-level, it is recommended to discuss few more device-level keying mechanisms in emerging photonic realm as well. Key generation using photonic devices appears to enhance security in future systems. For example, there has been a recent effort [a] that discusses a novel key generation mechanism in photonic links.

[a] S. V. R. Chittamuru, et al., "SOTERIA: exploiting process variations to enhance hardware security with photonic NoC architectures," in Proc. Of IEEE/ACM Design Automation Conference (DAC), June. 2018.

We added explanation of the device-level approach to enhance the security in Section 1.3.

Thank you very much again for your valuable comments.

Reviewer 2 Report

Overall I thought the paper provided a good discussion on re-keying scheme, which is side channel attack (SCA) resistant especially when the encryption key is refreshed with a randomness. The organization of the paper can be improved. In addition, more information needs to be presented in the paper to improve understanding of the paper.

Big items:

-It is recommended to add experimental section in this paper which compares proposed security enhancement scheme with previously proposed re-keying mechanisms to appreciate authors contribution.

-Section 7: This section appears concise, it is recommended to quantify the overheads of the proposed rekeying scheme for these applications. In addition, it is even better compare and contrast this scheme overhead with other schemes as well.   

-Throughout this paper there are many single line paragraphs. To improve formatting of the paper, my suggestion is to either make these single line sentences into subheadings or merge them to next paragraph.       

-In Section 5.2, encryption and decryption schemes are presented as bullets. In current form these bullets are not clearly understandable. It is recommended to convert these bullets into algorithms such that a reader understand these schemes clearly.      

Small Things:

-Page 2 of 13: Section 1.3: This section starts with a single sentence paragraph, which is not recommended. It is suggested to merge this paragraph to next paragraph.

-Page 2 of 13: Section 1.3, the literature survey in this section can be improved. As this paper presents keying mechanisms at both device- and system-level, it is recommended to discuss few more device-level keying mechanisms in emerging photonic realm as well. Key generation using photonic devices appears to enhance security in future systems. For example, there has been a recent effort [a] that discusses a novel key generation mechanism in photonic links.

[a] S. V. R. Chittamuru, et al., "SOTERIA: exploiting process variations to enhance hardware security with photonic NoC architectures," in Proc. Of IEEE/ACM Design Automation Conference (DAC), June. 2018.

Author Response

(The authors gave the same response as above.)

Round 2

Reviewer 1 Report

The improvements in the paper are significant. Now it is clearer. Just a pair of minor things.

Line 49: "iinto"

Figure 9: It is not linked in the text.

Author Response

Reply Letter

Dear Editor and Reviewers,

We sincerely appreciate your invaluable comments on our manuscript. We revised our manuscript again in considering your comments as follows.

We added our reply in-line with the blue color.

Thank you again and best regards,

Yuichi Komano and Shoichi Hirose

Reviewer 1

Comments and Suggestions for Authors

Line 49: "iinto"

We corrected it.

Figure 9: It is not linked in the text.

We cited the figure, at the line 211.

Thank you very much again for your valuable comments.

Reviewer 2

Comments and Suggestions for Authors

The authors addressed all my concerns and I recommend to accept it paper with no further changes.

We changed our manuscript for the comments from the reviewer 1 and in correction of types (ciphetext -> ciphertext).

Thank you very much again for your valuable comments.

Reviewer 2 Report

The authors addressed all my concerns and I recommend to accept it paper with no further changes.

Author Response

(The authors gave the same response as above.)
